# Redox-Regulation of α-Globin in Vascular Physiology

**DOI:** 10.3390/antiox11010159

**Published:** 2022-01-14

**Authors:** Laurent Kiger, Julia Keith, Abdullah Freiwan, Alfonso G. Fernandez, Heather Tillman, Brant E. Isakson, Mitchell J. Weiss, Christophe Lechauve

**Affiliations:** 1Inserm U955, Institut Mondor de Recherche Biomédicale, University Paris Est Creteil, 94017 Créteil, France; laurent.kiger@inserm.fr; 2Department of Hematology, St. Jude Children’s Research Hospital, Memphis, TN 38105, USA; Julia.Keith@STJUDE.ORG (J.K.); Alfonso.Fernandez@STJUDE.ORG (A.G.F.); Mitch.Weiss@STJUDE.ORG (M.J.W.); 3Department of Bone Marrow Transplantation and Cellular Therapy, St. Jude Children’s Research Hospital, Memphis, TN 38105, USA; Abdullah.Freiwan@STJUDE.ORG; 4Department of Pathology, St. Jude Children’s Research Hospital, Memphis, TN 38105, USA; Heather.Tillman@STJUDE.ORG; 5Robert M. Berne Cardiovascular Research Center, University of Virginia School of Medicine, Charlottesville, VA 22908, USA; brant@virginia.edu

**Keywords:** α-globin, endothelial nitric oxide synthase (eNOS), cytochrome, redox system, arteries, blood pressure

## Abstract

Interest in the structure, function, and evolutionary relations of circulating and intracellular globins dates back more than 60 years to the first determination of the three-dimensional structure of these proteins. Non-erythrocytic globins have been implicated in circulatory control through reactions that couple nitric oxide (NO) signaling with cellular oxygen availability and redox status. Small artery endothelial cells (ECs) express free α-globin, which causes vasoconstriction by degrading NO. This reaction converts reduced (Fe^2+^) α-globin to the oxidized (Fe^3+^) form, which is unstable, cytotoxic, and unable to degrade NO. Therefore, (Fe^3+^) α-globin must be stabilized and recycled to (Fe^2+^) α-globin to reinitiate the catalytic cycle. The molecular chaperone α-hemoglobin-stabilizing protein (AHSP) binds (Fe^3+^) α-globin to inhibit its degradation and facilitate its reduction. The mechanisms that reduce (Fe^3+^) α-globin in ECs are unknown, although endothelial nitric oxide synthase (eNOS) and cytochrome *b*_5_ reductase (CyB5R3) with cytochrome *b*_5_ type A (CyB5a) can reduce (Fe^3+^) α-globin in solution. Here, we examine the expression and cellular localization of eNOS, CyB5a, and CyB5R3 in mouse arterial ECs and show that α-globin can be reduced by either of two independent redox systems, CyB5R3/CyB5a and eNOS. Together, our findings provide new insights into the regulation of blood vessel contractility.

## 1. Introduction

The globin protein superfamily shares an active motif conserved across eukaryotes, bacteria, and archaea [1,2,3]. These proteins present a conserved three-dimensional structure, the “globin fold” [4]. Globin-family proteins contain a central heme prosthetic group that mediates numerous biological functions related to the transport or metabolism of molecular oxygen (O_2_), carbon monoxide (CO), or nitric oxide (NO). Hemoglobin and myoglobin contain pentacoordinate heme and are involved primarily in O_2_ transport and storage [5]. In contrast, neuroglobin and cytoglobin contain hexacoordinate heme, in which the sixth coordinate position of the central iron atom is bound by a globin amino acid that competes for external ligand binding [6]. These globins, particularly cytoglobin, may serve as reversible ligand carriers and may also participate in redox reactions, but their biological functions are currently unclear [7]. Each member of the globin family of proteins has a distinct pattern of tissue expression, subcellular localization, and affinity for external ligands, including O_2_, carbon dioxide (CO_2_), and NO [8,9]. Globin proteins in the cells of the vascular wall can regulate vasodilatory NO signaling between endothelium and smooth muscle, inactivating and detoxifying NO by transforming it to nitrate (NO_3_^−^) or reducing nitrite (NO_2_^−^) to active NO under hypoxic conditions to enable autoregulation of tissue perfusion by smooth muscle [9].

Recently, α-globin and cytoglobin have been identified in arteriolar endothelial cells (ECs) and smooth muscle cells (SMCs), respectively, and have been noted to cause small artery vasoconstriction [10,11,12,13,14] by degrading the vasodilator NO in a redox reaction termed dioxygenation [NO + O_2_ (Fe^2+^) globin → NO_3_^−^ + (Fe^3+^) globin] [13,15,16,17]. Moreover, studies of individuals with α-thalassemia demonstrated increased flow-mediated dilation [18]. The increased vasodilation may be explained by the loss of α-globin genes resulting in lower α-globin levels in the vasculature, as none of the patients presented with anemia or hemolysis. Nitric oxide generated in blood vessel ECs by endothelial nitric oxide synthase (eNOS) diffuses into adjacent vascular SMCs (VSMCs) to induce their relaxation by activating soluble guanylyl cyclase [19,20,21]. Two identical eNOS monomers can associate to form a dimeric active enzyme; each monomer includes a C-terminal reductase domain and an N-terminal oxygenase domain (the heme domain). The reductase domain contains two subdomains, each with its own flavin nucleotide prosthetic group: flavin mononucleotide (FMN) or flavin adenine dinucleotide (FAD). The reductase domain is connected to the oxygenase domain by a flexible linker containing a calmodulin (CaM) binding sequence. Calmodulin with associated Ca^2+^ regulates electron shuttling between the reductase and oxygenase domains. When eNOS is in the dimeric state, the heme group of one monomer receives electrons from the FMN domain of the other monomer. The oxygenase domain contains a tetrahydrobiopterin (BH4) prosthetic group that supports electron delivery to the heme group and enhances its affinity for the substrate l-arginine [22]. The catalytic activity mediated by the oxygenase domain uses l-arginine and O_2_ as substrates for generating NO, with nicotinamide adenine dinucleotide phosphate (NADPH) and CaM being essential cofactors [23].

Globin heme iron can bind ligands reversibly or participate in chemical reactions via electron transfer. Accordingly, the iron exists mainly in one of two interconvertible redox states—oxidized (Fe^3+^) or reduced (Fe^2+^)—depending on the direction of electron transfer. In reactions in which globin is used as an electron donor, including dioxygenation, heme iron is converted from Fe^2+^ to Fe^3+^ [13,16,17,24]. To participate in subsequent reactions, oxidized (Fe^3+^) globin must be recycled to the reduced (Fe^2+^) form. In erythrocytes, the major globin reductase system consists of soluble cytochrome *b*_5_ type A (CyB5a) and NADH–cytochrome *b*_5_ reductase 3 (CyB5R3) [25,26]. Specifically, CyB5R3 transfers electrons to CyB5a, which then donates electrons to oxidized hemoglobin. Previous studies have demonstrated that CyB5R3 participates in α-globin reduction in ECs [11], although whether this occurs through CyB5a or via other mechanisms is unknown.

Unlike erythrocytes, ECs express α-globin, but not β-globin [11,12]. Free α-globin rapidly autoxidizes and denatures. The molecular chaperone α-hemoglobin-stabilizing protein (AHSP) protects free α-globin by preferentially binding to the (Fe^3+^) form and rapidly inducing the formation of a stable hexacoordinate state that limits further chemical reactions with reactive oxygen and nitrogen oxide species [27]. α-Globin can bind either eNOS or AHSP, with the latter being favored when heme iron is oxidized [11,12,28]. In solution, (Fe^3+^) α-globin–AHSP can be reduced by CyB5R3 + CyB5a or by the reductase domain of eNOS. As free α-globin is unstable, we have proposed that most or all (Fe^3+^) α-globin in ECs is bound to AHSP, which facilitates its reduction by cellular enzymes, its release, and its transfer to eNOS to participate in dioxygenation [12]. According to this model, α-globin is stabilized by AHSP, and its availability to degrade NO is regulated by the cellular redox status. The present study investigated the model further by examining (Fe^3+^) α-globin reductase systems in ECs.

## 2. Materials and Methods

### 2.1. Mice

Mouse experiments were conducted with 6- to 24-week-old littermates. Animal care and experimental protocols were approved by the Institutional Animal Care and Use Committee at St. Jude Children’s Research Hospital (protocol number: 579-100641-03/20 approved 10 March 2020, for 3 years). The study proposal was reviewed and approved by the Institutional Review Board at St. Jude Children’s Research Hospital.

### 2.2. Antibodies

The source, application, concentration, and vendor of each antibody are listed in Appendix A.

### 2.3. Blood Vessel Isolation

Mice were euthanized by CO_2_ asphyxiation. Thoracodorsal arteries (TDAs) and mesenteric arteries were isolated by dissection then cannulated and washed with Krebs-HEPES buffer supplemented with 1% BSA to remove red blood cells (RBCs).

### 2.4. Histology, Immune Staining, and Detection

For standard histologic analysis, mice were anesthetized, the arteries were exposed by dissection, and the animals were perfused with 10% formalin. All formalin-fixed, paraffin-embedded tissues were sectioned at 4 μm, and the sections were mounted and dried at 60 °C for 20 min. Immunohistochemical and singleplex immunofluorescence (IF) labeling methods were tested and compared to determine the antibody titrations for each protein marker to be used in the multiplex IF studies. All tissue sections used for IF labeling underwent antigen retrieval in prediluted Cell Conditioning Solution (CC1) (Ventana Medical Systems, Indianapolis, IN, USA), then the primary antibodies were serially applied using the U DISCOVERY 5-Plex IF procedure and the appropriate kits (all from Ventana Medical Systems) as described previously [29]. Coverslips were mounted on slides with ProLong Gold Antifade Mountant containing DAPI (Thermo Fisher Scientific, Waltham, WA, USA, cat. no. P36935). Arterial sections were visualized with a confocal laser scanning microscope (Leica, Wetzlar, Germany, SP8), using an HCPL APO20x/0.75 objective.

### 2.5. RNA Extraction and RT-qPCR

Total RNA was extracted from arteries as described previously [30]. cDNA was synthesized using an iScript cDNA Synthesis Kit (BioRad), then RT-qPCR analysis was performed using SYBR Green (Applied Biosystems, Foster City, USA, cat. no. 4334973). Gene-specific primers (Appendix A) were designed using Primer3 software (Howard Hughes Medical Institute, Cambridge, MA, USA). The absence of DNA was confirmed by subjecting 1 ng of each RNA preparation to RT-qPCR with specific primers. RT-qPCR reactions were performed in duplicate. To determine the relative mRNA amounts for the genes studied, the comparative ΔΔCt method with β-actin as the normalizing gene was employed.

### 2.6. Reduction of Oxidized (Fe^3+^) α-Globin Bound to AHSP

Oxygenated (Fe^2+^) α-globin (from the Weiss Laboratory [31]) (150 μM) was incubated with recombinant human AHSP (from the Weiss Laboratory [31]) (150 μM) at 25 °C for 10 min. The protein complex was then oxidized by adding potassium ferricyanide (to a 1.2 molar excess) under deoxy conditions. After incubation for 30 min, the ferricyanide was removed by Sephadex G-25 (GE Healthcare Life Sciences, Chicago, IL, USA) chromatography. The experiments were performed under anaerobic conditions with N_2_, O_2_, or CO. All reactions with CyB5R3 (Novus Biologicals, Centennial, CO, USA, cat. no. NBP1-78879) and/or CyB5a were performed at 25 °C, pH 7.4, in 50 mM potassium phosphate, 50 mM NaCl, 0.8 mM EDTA, with 100 units of bovine catalase (Sigma-Aldrich, Saint-Louis, MO, USA, cat. no. C40) and 50 units of superoxide dismutase (Sigma-Aldrich, Saint-Louis, USA, cat. no. S7571) being added to this system. All reactions with eNOS were performed at 25 °C, pH 7.4, in 50 mM potassium phosphate, 50 mM NaCl, 0.8 mM EDTA, 1.4 mM CaCl_2_, 3 µM CaM (Sigma-Aldrich, Saint-Louis, USA, cat. no. C4874), 50 µM BH4 (Sigma-Aldrich, Saint-Louis, USA, cat. no. T4425), plus 100 units of bovine liver catalase, with or without 0.2 mM L-arginine as specified. Five units of glucose oxidase and 1 mM β-D(+)-glucose were added to scavenge the O_2_ in order to fulfill the requirement for anaerobic conditions. Reduced β-NADPH (50 µM) and β-NADH (100 µM) were ordered from Sigma-Aldrich Saint-Louis, USA, cat. nos. N1630 and N4505, respectively. The concentrations of eNOS and (Fe^3+^) α-globin–AHSP were both 1 µM.

The O_2_ scavenging system was tested with O_2_ (Fe^2+^) α-globin diluted in a deoxygenated buffer after the addition of 10–20 µM O_2_ from an air-equilibrated buffer; the O_2_ (Fe^2+^) α-globin deoxygenation was completed within 1 min. To obtain a baseline recording, a sample of the buffer, deoxygenated or equilibrated under CO, was injected with a Hamilton syringe, through a rubber cap, into an O_2_-free sealed quartz optical cuvette containing all the reaction components. Highly concentrated stock solutions were used to avoid the need to add more than 5 µL to a final reaction volume of 500 µL (with glucose being added last). An initial spectrum was recorded with a Jasco V-670 spectrophotometer after adding 1 μM recombinant soluble eNOS to the O_2_-free reaction buffer. After reduced NADPH was added to a final concentration of 50 µM, the reduction kinetics of eNOS with or without CO were recorded with an HP 8453 diode-array spectrophotometer. After the readings reached a plateau, the spectrum of the partially reduced eNOS was recorded, and a calculated spectrum was used for data analysis after subtracting the previous baseline reading and the contribution of NADPH absorption. Finally, the (Fe^3+^) α-globin–AHSP reduction kinetics with and without CO were further monitored by diluting samples in partially reduced eNOS, and the final absorption spectrum was analyzed by following the same protocol. Other kinetics were measured by adding eNOS to the same buffer conditions, i.e., containing the O_2_ scavenging system, (Fe^3+^) α-globin–AHSP, and NADPH.

Each condition was duplicated for each rate measured to evaluate the variability in the fits obtained using SigmaPlot 10.

### 2.7. Statistical Analysis

Data represent the mean ± SEM. Statistical analyses were performed using Prism 6.0 (GraphPad Software, San Diego, CA, USA).

## 3. Results

### 3.1. α-Globin, CyB5R3/CyB5a, and eNOS Are Expressed in Arterioles

To characterize the expression of α-globin, CyB5R3/CyB5a, eNOS, and cytoglobin in the thoracodorsal artery (TDA) (non-resistant) and mesenteric (resistant) arteries, we determined the mRNA steady-state levels in normal arteries by RT-qPCR analysis. The steady-state levels of α-globin (*Hba1* and *Hba2*) and eNOS (*Nos3*) mRNA expression are equivalent in the TDA and mesenteric arterioles (Figure 1A) (Appendix A). Similarly, CyB5a and CyB5R3 are also expressed in equivalent steady-state levels in the TDA and mesenteric arterioles. In contrast, cytoglobin, characterized as being expressed in SMCs [13], is expressed at a level 10 times that of α-globin in both of these tissues (Figure 1A). As β-globin was previously not detected in arteriolar extracts [12,28], we assume that the *Hbb* mRNA detected came from erythroid cell contamination.

Immunofluorescence multiplex staining of TDA and mesenteric arteriole sections from 6-month-old mice detected α-globin and eNOS, which is consistent with our previous reports (Figure 1B,C, white arrows) [11,12]. In ECs, the distribution of CyB5a and CyB5R3 overlapped that of α-globin, consistent with the co-localization of the three proteins. α-Globin and eNOS signals were detected almost exclusively in arteriolar ECs, whereas CyB5a and CyB5R3 were detected in both ECs and VSMCs.

Another cytochrome, cytochrome *b*_5_ type B (CyB5b), has been suggested to be a donor of electrons to cytoglobin [32], so we investigated its expression and distribution and found that it is also expressed in arteriolar ECs and VSMCs. Therefore, CyB5b is indeed a potential electron donor, although the mitochondrial localization of this cytochrome [33] should limit its bimolecular interactions with α-globin and cytoglobin.

### 3.2. eNOS Reductase and Oxygenase Domain Activities

The discovery that eNOS is an α-globin-reducing agent was unanticipated [12]. To study this reduction reaction further, we tested its dependence on eNOS cofactors required for NO production. We first studied the effect of the cofactors BH4 and CaM/Ca^2+^ ± L-arginine on the reduction of the oxygenase domain of one monomer by the FMN domain of the other monomer in the functional eNOS dimer (Figure 2 and Table 1). We estimated the reduction rate by measuring the CO binding to the (Fe^2+^) eNOS oxygenase domain (Appendix A). CO binds rapidly and with high affinity to the (Fe^2+^) heme domain; the limiting factor in this condition is the rate of reduction of the oxygenase domain by the reductase domain (electrons flow through the flavin domains until the final interdomain transfer toward the oxygenase domain). The reduction of the oxygenase domain was monophasic, and the presence of L-arginine had only a low influence on these rates (Table 1). The rate of reduction, 0.003 s^−1^, is consistent with what has been reported previously [34]. In the steady state, with the cofactors (BH4 and CaM/Ca^2+^ ± l-arginine) present, we consistently observed a partial (~60%) reduction of the oxygenase domain of eNOS, as compared to the complete reduction observed with sodium dithionite (Figure 2). In the presence of cofactors (BH4 and CaM/Ca^2+^), eNOS exhibited a slow rate of reduction of the oxygenase domain, probably to avoid, especially in the presence of O_2_, the catabolism of NO, its own catalysis product (Figure 2 and Table 1). At equilibrium under these conditions (with approximately 60% of the heme reduced), we assume that only a single oxygenase domain of the eNOS coupled dimeric protein is being reduced. We anticipate the involvement of an additional mechanism of negative cooperativity in the reduction of a second oxygenase domain. The kinetic features may be also explained, at least in part, by the electron repartition between the FAD reductase subdomain and both reduced conformations (semiquinone and hydroquinone) of the FMN subdomain.

### 3.3. Reduction of α-Globin by eNOS

After establishing that full-length eNOS protein was dimeric and functionally active, we measured the reduction of (Fe^3+^) α-globin–AHSP by eNOS under anaerobic conditions in the presence of BH4 and CaM/Ca^2+^ and in the presence or absence of L-arginine and CO (Figure 3A and Table 1). We limited the use of L-arginine in our experiments to avoid NO production and associated redox side reactions in the presence of O_2_ (even if we used an O_2_ scavenging system). We compared the results to the eNOS oxygenase reduction rate to evaluate a potential interdomain electron transfer from reduced eNOS-deoxygenated (Fe^2+^) heme. Full reduction was completed within a few tens of seconds with 1 µM protein concentrations. When CO was added, we observed fully saturated α-globin (100% heme reduced) (Figure 3B: the dotted gray line compared to the solid black line at 420 nm) and approximately 60% saturation of the reduced eNOS oxygenase domain (Figure 3B: the dotted gray line compared to the solid black line at 445 nm). When we added NADPH after mixing eNOS and (Fe^3+^) α-globin–AHSP under CO to measure the competition for electron capture between the two heme domains, α-globin was reduced well before the eNOS oxygenase domain, in accordance with the reduction rates reported in Table 1 (data not shown). The reduction rates for the eNOS oxygenase domain were slower than those for the reduction of (Fe^3+^) α-globin–AHSP by at least an order of magnitude (Table 1), which indicates that α-globin reduction can compete in the range of nanomolar concentrations, even with BH4 and CaM/Ca^2+^ bound to the eNOS dimeric structure. Adding CO did not decrease the rate of electron transfer, probably because the intermolecular pathway for α-globin reduction passes through the eNOS reductase domain, with reduced FMN serving as the final electron donor. The kinetics of α-globin reduction by eNOS were monophasic (Figure 3A, Appendix A, and Table 1).

### 3.4. Reduction of α-Globin by the CyB5R3/CyB5a System

CyB5a and CyB5R3 are proteins with methemoglobin reductase activity. We determined the ability of CyB5R3 to reduce (Fe^3+^) α-globin–AHSP in the presence or absence of its partner protein CyB5a. Experiments conducted in the absence of CyB5a showed that CyB5R3 in the presence of NADH cannot reduce α-globin directly. The rate of CyB5R3 reduction (without CyB5a) was too low for this reduction to be involved in a turnover reaction for the NO dioxygenase activity of α-globin in ECs (Table 1). Therefore, CyB5R3 requires CYB5a as an intermediate electron acceptor that in turn reduces α-globin (Table 1). The major difference in redox potential between CyB5R3 (−265 mV) and CyB5a (+20 mV) favors electron transfer [35], but CyB5a and (Fe^3+^) α-globin are required to form a complex whose bonding is principally determined by complementary charge interactions between acidic groups of CyB5a and basic groups of α-globin [36]. α-Globin as a monomer and as incorporated in the hemoglobin tetramer exhibits the same reactivity with CyB5a [37]. Considering the slower consensus rate of 0.04 s^−1^, the eNOS reductase domain reduced (Fe^3+^) α-globin at a rate at least an order of magnitude faster than that observed with a CyB5R3/CyB5a system (Table 1).

## 4. Discussion

The globin protein superfamily is an ancient motif, with related proteins being found in eukaryotes, bacteria, and archaea [1,38]. Globin diversity can be explained by the adaptive values of the members of the globin gene family in vertebrates, all produced by gene duplication and sequence diversification, based on taxonomic distribution, cellular localization, and knowledge regarding their functions [7]. Primordial globin proteins are believed to have arisen in low-O_2_ environments as O_2_ sensors and/or enzymatic NO scavengers before evolving into RBC O_2_ transporters [39], and the evolution of globin genes over time provides a context for understanding the diverse roles of globins in vascular physiology. Globins carry out their diverse functions through reversible binding of O_2_ for transport and storage to protect cells against reactive oxygen species, NO scavenging, signaling in O_2_-dependent metabolic pathways, and possibly other functions involving ligand or electron transfer [40]. Globin proteins in the cells of the vascular wall can regulate vasodilatory NO signaling between endothelium and smooth muscle [10,11,12,13,14,41,42]. Each globin has unique properties with respect to regulating NO/nitrite/nitrate and O_2_ affinity [9]; however, to provide a continuous reserve of NO scavenging (through dioxygenation, in normoxic conditions) or production (nitrite reductase, in hypoxic conditions), oxidized (Fe^3+^) globin needs to be recycled to the (Fe^2+^) globin reduced state. In the myoendothelial junctions (MEJs) of ECs, oxygenated (Fe^2+^) α-globin bound to eNOS promotes vasoconstriction by degrading locally produced NO to generate NO_3_^−^ and oxidized (Fe^3+^) α-globin, which is unable to degrade NO further [11,28]. Previously, we and others have shown that α-globin can be reduced by the CyB5R3/CyB5a or eNOS systems [12,43,44]. Here, we have confirmed that these two systems are present in ECs and co-localize with α-globin. In addition, we have performed biochemical studies to characterize their reductase rates and cofactor requirements under various physiologically relevant conditions (Table 1).

Electron transfer from the CyB5R3, CyB5a, and eNOS reductase domains is thermodynamically favorable, because the redox potentials are below those of free or AHSP-bound α-globin [45,46], although our experiments revealed that α-globin cannot be reduced directly by CyB5R3. The surface at the entrance to the α-globin heme pocket is positively charged owing to the proximity of several lysine residues (Lys 60/Lys 61/Lys 90) [47]. This structural arrangement is compatible with the formation of a transient bimolecular contact between α-globin and the negatively charged eNOS FMN subdomain or CyB5a [48,49,50,51]. The positively charged surface of the FAD subdomain or CyB5R3 limits the formation of a favorable interface with α-globin and prevents the close flavin/heme orientation, and it should, therefore, limit efficient electron transfer [48,49,50,51]. Nevertheless, CYB5R3 has been found to contribute to the reduction of α-globin in ECs to mediate NO scavenging at MEJs [11]. Consequently, the reduction of free or AHSP-bound (Fe^3+^) α-globin by CyB5R3 in ECs would require sufficient CyB5a to serve as the final electron donor, similar to the requirements for both proteins for the reduction of oxidized hemoglobin in RBCs [12,43,44]. Here, we have shown that CyB5a and CyB5R3 are both expressed in TDA and mesenteric arteriole ECs (Figure 1), acting together to reduce α-globin. Both proteins are also expressed in VSMCs, in which cytoglobin has recently been shown to degrade NO by dioxygenation [11,12,13].

We have shown in this study that eNOS (and its cofactors) can reduce (Fe^3+^) α-globin–AHSP at a rate that is an order of magnitude faster than that observed with the CyB5R3/CyB5a system (Table 1). The interaction domain of α-globin with eNOS (or CyB5a), to enable electron transfers, is not limited by the binding of AHSP. We previously determined that eNOS can serve as a direct electron donor to (Fe^3+^) α-globin, highlighting a mechanism for globin protein reduction that probably contributes to EC physiology [12]. In ECs, Ca^2+^/CaM-bound eNOS is enzymatically active and requires NADPH, BH4, L-arginine, and molecular O_2_ as cofactors to produce NO [52]. Oxidized (Fe^3+^) eNOS generated in this reaction is converted to reduced (Fe^2+^) heme via electron transfer from the reductase domain of the opposite monomer in the functional eNOS dimer. Electron-transfer reactions involve a transfer from NADPH to the FAD subdomain, which then passes electrons sequentially to the FMN subdomain of the reductase subunit. Once the FMN domain receives two electrons (resulting in the semiquinone conformation), the redox potential is low enough that it can transfer one electron to the eNOS oxygenase domain or to another heme-protein acceptor such as α-globin or, as previously described, cytochrome *c* [52]. Interestingly, the rate of reduction of the α-globin heme by eNOS was faster than that of the intramolecular reduction of the eNOS heme by its own reductase. This new finding suggests that eNOS can participate in cellular electron-transfer processes in an active dimeric configuration (intramolecularly/intermolecularly) (as in the current study) or in an inactive monomeric uncoupled configuration (intermolecularly) [12] and can reduce α-globin as a heterologous electron acceptor.

Catabolism of NO by oxygenated (Fe^2+^) globins is limited only by the diffusion of a diatomic ligand into the heme pocket [53], and the presence of two reducing systems is probably sufficient to bring one electron at a time for the turnover cycle. Nevertheless, post-translational modifications of eNOS, including phosphorylation of Ser1177, have been characterized as modulating the reduction rate of the oxygenase domain as well as increasing NO synthesis [54]. These modifications may be suitable for modulating NO availability, which would then outcompete α-globin scavenging [55,56]. The NO metabolism could also be differentially influenced by other variables, such as the protein abundance, the localization in the subdomain (relocalization to the MEJ), or protein activation/inactivation based on redox compartmentalization in the cellular environment. Regardless, it is interesting that eNOS itself, through its reductase domain and in the presence or absence of cofactors necessary for NO synthesis, is intrinsically the most efficient reducing system for fine-tuning the regulation of cellular NO metabolism.

Our findings support the conclusion that NO scavenging in ECs is regulated by redox-dependent shuttling of α-globin between eNOS and AHSP, which, in turn, is regulated by EC reductase systems (Figure 4). In the MEJ, oxygenated ferrous (Fe^2+^) α-globin bound to eNOS degrades locally produced NO via dioxygenation, generating NO_3_^−^ and deoxygenated ferric (Fe^3+^) α-globin, which can no longer degrade NO. eNOS-bound (Fe^3+^) α-globin may be reduced to (Fe^2+^) α-globin by eNOS directly or by the associated CyB5R3/CyB5a redox system. Alternatively, (Fe^3+^) α-globin may be transferred to AHSP, which has an approximately 100-fold greater affinity for this redox form when compared to (Fe^2+^) α-globin [57,58]. Previous studies have shown that AHSP stabilizes bound (Fe^3+^) α-globin by converting it to a hexacoordinate structure in which heme iron is bound on either side of the planar ring by histidine α-globin residues located at α-helical positions F8 and E7 [27,59]. Under reducing conditions, CyB5R3/CyB5a-mediated conversion of (Fe^3+^) α-globin to (Fe^2+^) α-globin would favor the release of α-globin from AHSP and its transfer to eNOS in ECs (or to β-globin in RBCs [57,60,61]). Therefore, we theorize that AHSP functions in ECs to sequester and stabilize non–eNOS-bound α-globin, particularly the unstable and potentially damaging oxidized (Fe^3+^) form. Although the kinetics of α-globin binding to eNOS are unknown, our model predicts that equilibria favor the binding of (Fe^2+^) α-globin to eNOS over (Fe^3+^) α-globin to AHSP. Under conditions that promote heme oxidation, such as when there is a high rate of NO degradation by α-globin, low O_2_ tension, or acidosis [62,63], AHSP could sequester (Fe^3+^) α-globin in an inert, but stable, state to minimize NO degradation, promote vasodilation, and enhance tissue O_2_ delivery. Further supporting this model, binding of AHSP to (Fe^3+^) α-globin facilitates its enzymatic or chemical reduction to (Fe^2+^) α-globin and the subsequent dissociation of the complex [62].

## 5. Conclusions

Overall, our studies have further highlighted the importance of α-globin in endothelium and the fine-tuned control of NO regulation in the arteriolar vascular wall. The reduced form of α-globin is required for oxygen binding and inhibitory NO deoxygenation reactions. Our model predicts that NO diffusion or scavenging in ECs is regulated by redox-dependent shuttling of α-globin. We have shown that two independent redox systems, CyB5R3/CyB5a or eNOS, can reduce α-globin under various physiologically relevant conditions, though further studies are required to validate this model to fully appreciate the implications of our findings in mice and humans.

## Figures and Tables

**Figure 1 antioxidants-11-00159-f001:**
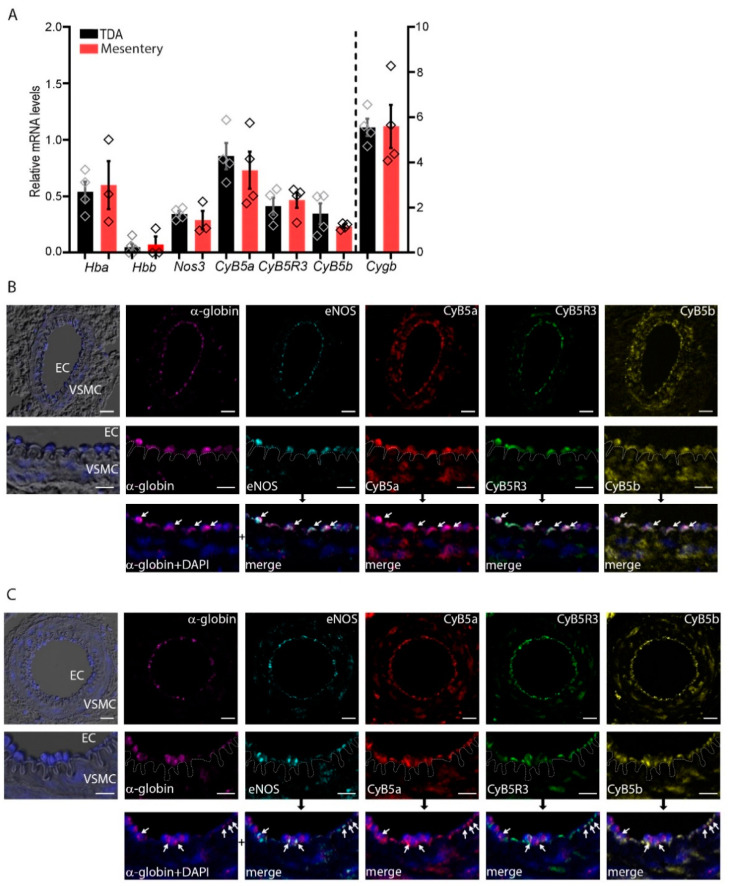
α-Globin and redox systems are co-expressed in small arteries. (**A**) mRNA levels in the thoracodorsal artery (TDA) and mesenteric arteries. The bar chart shows data from three or four 6-month-old mice. (**B**,**C**) Indirect immunofluorescence multiplexing staining for α-globin, endothelial nitric oxide synthase (eNOS), CyB5a, CyB5b, and CyB5R3 (white arrows) in TDA (**B**) and mesenteric artery (**C**) preparations. The dashed line across the internal elastic lamina demarcates the ECs and VSMCs. Merges represent the overlap of “α-globin+DAPI” with other staining based on the black arrow upstream. DAPI-stained nuclei are shown in blue, in combination with differential interference contrast. Scale bars are equivalent to 10 μm and 20 μm (for the higher magnification).

**Figure 2 antioxidants-11-00159-f002:**
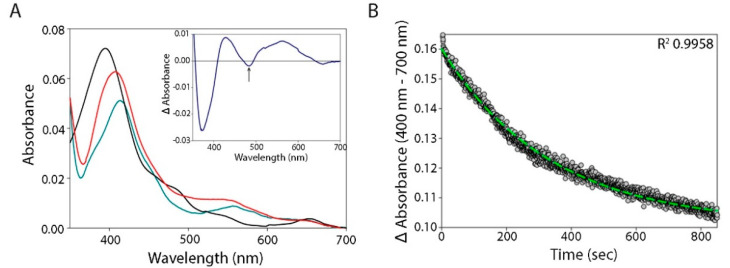
eNOS reduction. (**A**) Spectra for full-length Fe^3+^ eNOS (black line), after the addition of NADPH (red line) or with sodium dithionite (cyan line), under anaerobic conditions. The red spectrum is simulated as a combination of 40% ferric and 60% ferrous eNOS. The insert shows the differential spectrum for the oxygenase domain redox transition of the full-length eNOS after addition of NADPH, which also involves flavin reduction in the reductase domain (485 nm, black arrow). (**B**) eNOS reduction kinetics in the presence of cofactors and L-arginine after mixing NADPH under anaerobic conditions. The dashed lines represent the exponential rise fit with the associated determination coefficient (R^2^).

**Figure 3 antioxidants-11-00159-f003:**
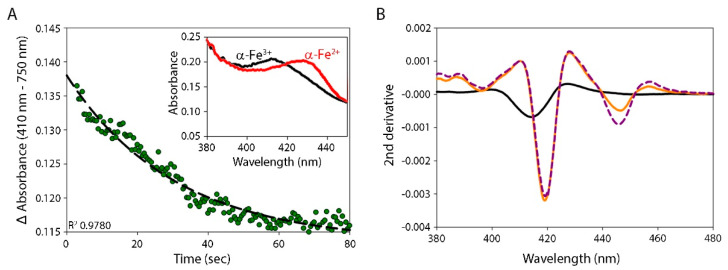
α-Globin reduction by eNOS. (**A**) α-Globin–AHSP reduction kinetics after mixing with eNOS pre-incubated under anaerobic conditions in the presence of NADPH (as presented in Figure 2A). The chosen detection wavelength of 410 nm is close to the isosbestic point of the eNOS redox transition (as presented in Figure 2B). The dashed line represents the exponential decay fit with the associated determination coefficient (R^2^). The insert shows the transition spectrum of the oxidized (black line) and reduced (red line) α-globin. (**B**) The second derivative of the eNOS/NADPH and α-globin–AHSP mixture after the completion of globin reduction under anaerobic conditions with CO. The (Fe^3+^) α-globin–AHSP second derivative is centered at 414 nm, which is characteristic of bis-histidyl hexacoordination (black solid line). After reduction, the (Fe^2+^) α-globin–CO is revealed with the minima of its second derivative centered at 419 nm, whereas the (Fe^2+^) eNOS–CO is revealed with the minima centered at 445 nm (orange line). Upon the addition of dithionite, the amplitude of the (Fe^2+^) eNOS–CO second derivative increases almost two-fold (purple dashed line) by comparison with that of (Fe^2+^) α-globin–CO, which does not change (orange line). This underlines the fact that (Fe^3+^) α-globin–AHSP is fully reduced by eNOS after a few tens of seconds, whereas eNOS is only partially reduced by approximately 50%. All reaction were performed at 25 °C.

**Figure 4 antioxidants-11-00159-f004:**
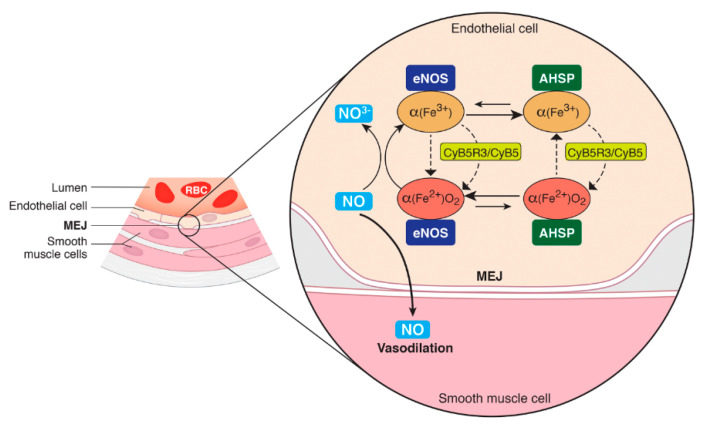
Model for regulation of NO metabolism by α-globin in the myoendothelial junction (MEJ) of vascular endothelial cells. After translation, newly formed apo–α-globin subunits bind to their partner subunit, AHSP, or, to a lesser extent, eNOS. AHSP can bind to both reduced (Fe^2+^) and oxidized (Fe^3+^) α-globin. In the reduced complex, the heme group is coordinated with the bound O_2_. O_2_ (Fe^2+^) α-globin interacts with eNOS and degrades NO via dioxygenation or binds AHSP and rapidly becomes oxidized. The oxidized α-globin is catalytically inert, and it is this function of AHSP that protects cells from oxidative damage. However, the specific binding of α-globin by AHSP or eNOS prevents α-globin from precipitating, allowing more time for α-globin to be reduced by eNOS directly or by the CyB5R3/CyB5 system and become functional.

**Table 1 antioxidants-11-00159-t001:** Reduction rates for α-globin–AHSP and eNOS.

	Enzymatic System	Reduction Rate (s^−1^) ± SD	Conditions
α-Globin–AHSP (1 µM)	CyB5R3/CyB5a/NADH (0.1 µM/1 µM/100 µM)	0.003 ± 0.00004	76 Torr CO
	CyB5R3/NADH (0.1 µM/100 µM)	0	76 Torr CO/10 Torr O_2_
	eNOS/NADPH/BH4/CaM/Ca^2+^ (1 µM/50 µM/50 µM/3 µM/1.4 mM)	0.04 ± 0.0007	76 Torr CO
	eNOS/NADPH/BH4/CaM/Ca^2+^/L-arginine (1 µM/50 µM/50 µM/3 µM/1.4 mM/0.2 mM)	0.05 ± 0.0019	Anaerobic
eNOS (1 µM)	NADPH/BH4/CaM/Ca^2+^ (50 µM/50 µM/3 µM/1.4 mM)	0.008± 0.00005	76 Torr CO
	NADPH/BH4//CaM/Ca^2+^/L-arginine (50 µM/50 µM/3 µM/1.4 mM/0.2 mM)	0.003 ± 0.00002	Anaerobic

Reduction of eNOS with its cofactors was initiated after addition of NADH. Reduction of α-globin–AHSP by CyB5R3 in the presence or absence of its partner protein CyB5a or by eNOS was initiated after addition of the globin to the enzymatic reaction components. All reaction were performed at 25 °C.

## Data Availability

The data is contained within the article or Appendix A.

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
