# Peer review of "Redox-Regulation of α-Globin in Vascular Physiology"

_antioxidants, 2022, doi:10.3390/antiox11010159_

Round 1

Reviewer 1 Report

α-globin and cytoglobin have been identified in arteriolar endothelial cells (ECs) and smooth muscle cells (SMCs), respectively, and have been noted to cause small artery vasoconstriction  by degrading the vasodilator NO in a redox reaction termed dioxygenation [NO + O2 (Fe2+) globin → NO3− + (Fe3+) globin]. This reaction converts reduced (Fe2+) α-globin to the oxidized (Fe3+) form, which is unstable, cytotoxic, and unable to degrade NO. Therefore, (Fe3+) α-globin must be stabilized and recycled to (Fe2+) α-globin to reinitiate the catalytic cycle.

 In this work, the authors proves that eNOS (and its cofactors) can reduce (Fe3+) α-globin–AHSP at a rate that is an order of magnitude faster than that observed with the CyB5R3/CyB5a system. The interaction domain of α-globin with eNOS (or CyB5a), to enable electron transfers, is not limited by the binding of AHSP.

This is an excellent work and interest the readers in this filed.

Please improve the Figure 2 and 3 with higher resolution for the readers understanding them easily. It is better to use the different colors in Figure 3 B to separate different states of samples. It suggest to change the font of X and Y axis in Figure 2 and 3 to larger size.  

Author Response

Reviewer 1:

Please improve the Figure 2 and 3 with higher resolution for the readers understanding them easily. It is better to use the different colors in Figure 3 B to separate different states of samples. It suggest to change the font of X and Y axis in Figure 2 and 3 to larger size. 

We have revised Figures 2 and 3 according to the suggestions of the reviewer.

Reviewer 2 Report

This paper entitled "Redox-regulation of alpha-globin in vascular physiology" extends interesting speculations from the last decade about a possible role of free alpha-globin proteins in the myoendothelial junctions of arterioles in controlling the production and function of nitric oxide.  This paper from the several laboratories that have published on this hypothesis recounts in some detail these ideas and adds data on the needed reduction of the alpha-globin species after postulated oxidation by eNOS produced NO, including stabilization of the met-alpha-globin by the well-characterized AHSP and reductive reactions with eNOS itself as well as the CyB5R3/CyB5a system.

  However, in view of the potential importance of this hypothesis if further validated, I find the paper lacking in many aspects that the authors could have easily included to provide further evidence for these ideas (i.e., to test the underlying hypothesis) as well as attempting to expand the paradigm with new results. The paper presents sophisticated analyses of the reductive processes of alpha globin with an uncharacterized eNOS preparation (said to be a gift from another laboratory) but these results - in an arbitrary in vitro system - advance the field to a very limited extent in view of the analyses that are needed to test this model.

  For example, why do we not see measurements of beta-globin mRNA (as well as alpha-globin) and what do the" relative" amounts of steady state mRNA in Figure 1A tell us and indeed why were these two types of arteries used?  I would think that analyses of endothelial cells as well as the VSMC sometimes used should also be included so we know in which cells these processes actually occur.  Similarly the quality of the photomicrographs Figs 1B and 1C in my copy are very poor and one cannot make out where the various labels are located in these preparations and what "co-localization" means.  In particular what is the evidence of these studies of whole arteries that these processes are occurring in myo-endothelial junctions as concluded in the Discussion and in the last Figure.  Can vascular preparations without MEJ be used to test this aspect of the model?

  Studies with and without L-arginine or NO destructive reagents to show an expected effect of NO generation by the eNOS on the oxidation state of the alpha-globin would also be important, as a complement to the kinetic studies presented here. I am also concerned about the fact that reduction only occurred to 60% and dithionite had to be added (with all sorts of side reactions) to the preparations as well as the fact that many of the kinetic measurements were done under anerobic conditions.  Why?

   Similarly, the references in a number of cases are in my opinion inappropriate. One example of such distortions is the statement: "Recently, studies from individuals with α-thalassemia presented an increased flow-mediated dilation and demonstrated a relationship between the absence of α-globin in the vessel wall and increased arterial vasodilation [18]."  This paper has no measurements of alpha-globin in the vessel wall and the small effects (which not "arterial vasodilatation") are with a single locus deletion (has the effect if any on alpha-globin levels in MEJ been determined?) and equally surprising there was no effect in that paper on levels of FMD of the alpha-globin "dose."

  A major revision of this manuscript with rigorous testing of the underlying hypotheses rather than publication of this note would be in the interest of the field in general as well as these authors and would help establish the extent of the validity of this interesting model.  

Author Response

Reviewer 2:

The paper presents sophisticated analyses of the reductive processes of alpha globin with an uncharacterized eNOS preparation (said to be a gift from another laboratory) but these results - in an arbitrary in vitro system - advance the field to a very limited extent in view of the analyses that are needed to test this model.

We received full-length eNOS protein from Drs. Poulos and Li (University of California, Irvine), both of whom are experts on nitric oxide synthase (NOS) proteins. We reasoned that it would be important to control for the functionality of the protein by using spectrophotometric analysis to measure the reduction of one oxygenase domain of one monomer by the FMN domain of the other monomer in the functional eNOS dimer, which occurs only in the presence of the cofactors BH4 and CaM/Ca2+ (Figure 2). In our previous studies, we did not control for these effects in the presence of these cofactors.

For example, why do we not see measurements of beta-globin mRNA (as well as alpha-globin) and what do the" relative" amounts of steady state mRNA in Figure 1A tell us and indeed why were these two types of arteries used? 

We have revised the figures and the manuscript according to the suggestions of this reviewer (lines 189-190). The relative mRNA amount for each gene studied was used to confirm the expression and relative proportions of the enzymatic partners. Our previous data were generated in TDAs which bare similar vaso-reactive properties to resistance arteries (e.g., minimal NO-mediated dilation and myoendothelial junctions) but are anatomically feed arteries for the spinotrapezius muscle. In this study we moved our studies to third order mesenteric arteries which are the gold standard for resistance arteries—the arteries that regulate total peripheral resistance component of blood pressure. This is vital because it immediately extends our findings to >1000 published works using these resistance arteries in which we can now begin to make comparisons of our findings.

I would think that analyses of endothelial cells as well as the VSMC sometimes used should also be included so we know in which cells these processes actually occur. 

This is an excellent point. This work has already been performed and described by our collaborators (PMID: 23123858; PMID: 25278292). It was found that α-globin is expressed in endothelial cells and is enriched at myoendothelial junctions, the cellular domain where endothelium and smooth muscle make contact.

Similarly, the quality of the photomicrographs Figs 1B and 1C in my copy are very poor and one cannot make out where the various labels are located in these preparations and what "co-localization" means.

The quality of the original imaging, obtained with a confocal laser scanning microscope, is actually very good. We have taken care to provide the journal with images of the highest-possible resolution for publication. We decided to delineate the endothelial and smooth muscle cells by using the internal elastic lamina from the differential interference contrast images. Please see the new Figures 1B and 1C.

We use the term co-localization to refer to the overlapping signals in multiplexed images.

In particular what is the evidence of these studies of whole arteries that these processes are occurring in myo-endothelial junctions as concluded in the Discussion and in the last Figure.  Can vascular preparations without MEJ be used to test this aspect of the model?

This is an excellent comment. We have revised the manuscript in response and have properly referred to the work that previously showed that these reactions occur at the myoendothelial junction (lines 318-321). Work by co-author Isakson has extensively demonstrated the presence of alpha globin in myoendothelial junctions of resistance arteries (TDAs, resistance arteries:  PMID 23123858, 30900949 and 25278292; as well as 32463112). Anatomically, myoendothelial junctions do not exist in carotid arteries, and neither does alpha globin. Indeed, cultured endothelial cells express very little alpha globin unless cellular contact can be initiated with smooth muscle (PMID: 23123858, 30900949). For this reason, vascular preparations where alpha globin is endogenously present, aren’t possible without introduction of plasmids, etc.

Studies with and without L-arginine or NO destructive reagents to show an expected effect of NO generation by the eNOS on the oxidation state of the alpha-globin would also be important, as a complement to the kinetic studies presented here. I am also concerned about the fact that reduction only occurred to 60% and dithionite had to be added (with all sorts of side reactions) to the preparations as well as the fact that many of the kinetic measurements were done under anerobic conditions.  Why?

We measured the reduction of α-globin by eNOS with or without L-arginine. We limited to use L-arginine in our experiments to avoid NO production and associated redox side reactions in presence of O2 (even if we used an O2 scavenging system). Moreover, as we mentioned in the text, phosphorylation of Ser1177 can modulate the reduction of heme.

We have also modified Figure 2 and now show the fully reduced deoxy spectrum, along with the kinetics of reduction of eNOS in the insert. We observed a similar partial reduction of eNOS in the presence of CO, as shown in Figure 3, and described on line 254: “When CO was added, we observed fully saturated α-globin (100% heme reduced) (Figure 3B: the dotted gray line compared to the solid black line at 420 nm) and approximately 50-60% saturation of the reduced eNOS oxygenase domain (Figure 3B: the dotted gray line compared to the solid black line at 445 nm).”

We performed the measurements under anerobic conditions to avoid interference by O2 and the formation of oxidative species that might interfere with the function and/or stability of the proteins.

Similarly, the references in a number of cases are in my opinion inappropriate. One example of such distortions is the statement: "Recently, studies from individuals with α-thalassemia presented an increased flow-mediated dilation and demonstrated a relationship between the absence of α-globin in the vessel wall and increased arterial vasodilation [18]."  This paper has no measurements of alpha-globin in the vessel wall and the small effects (which not "arterial vasodilatation") are with a single locus deletion (has the effect if any on alpha-globin levels in MEJ been determined?) and equally surprising there was no effect in that paper on levels of FMD of the alpha-globin "dose."

We understand the reviewer may have concerns about the conclusions from the referenced peer-reviewed manuscript, but it is highly relevant to the work we demonstrate in the paper. We don’t believe we shouldn’t cite this manuscript because they didn’t measure alpha globin from living human arteries where FMD was performed. The conclusions from that paper do fall in line with work we present here. However, to address this concern, we have revised the portion of our manuscript that refers to this published work to be more speculative about the physiologic explanation for this measurement. See lines 56-60.

Reviewer 3 Report

This manuscript covers an interesting topic that is highly relevant to vascular NO function. However, the majority of the findings presented herein have already been published by either this same group or by Dr. Stuehr’s.

Main points

The kinetics of eNOS heme reduction by NADPH were reported by Abu-Soud and Stuehr in JBC 2000; 275(23)17349. Similar to Abu-Sound et al, the authors report that eNOS heme reduction follows biphasic kinetics. Please provide representative raw traces and corresponding fits in Figure 2.

Using a stopped-flow apparatus at 10oC, Abu-Sound et al reported a k = 85s-1 for the fast phase of eNOS heme reduction by NADPH. Does this correspond to the same process reported in Table 1 to occur with a rate constant of 0.02 s-1? Why is there such a large discrepancy?

Please provide the spectrum for fully reduced eNOS in Figure 2A to better show the isosbestic point at 410nm.

Consistent with Abu-Sound et al, the presence or absence of L-Arg is stated to have no effect on eNOS heme reduction rates. However, neither the rate constants with and without Arg, nor a statistical analysis of this comparison are provided.

No information is provided regarding the number of replicates and standard deviations for the kinetic determinations, the r2 of the fits or any other statistical analysis.

The reduction rates for the reduction of alpha-globin by eNOS and by the CyB5R3/CyB5a system were reported by the authors in a 2018 paper in the Journal of Clinical Investigation. The only apparent difference between the original report and the present manuscript is that a previously unobserved rapid phase is now reported with a rate estimated at 0.5 s-1. Given that this observation seems to be the main difference between the 2018 paper and the present manuscript, the authors are advised to attempt a better characterization of this hyper-reductive rate by either working at lower temperatures or using a stopped-flow system.

The hyper-reductive rate is not apparent in figure 3A and the model used to fit the kinetic trace is not specified. Is this a biphasic or a monophasic model?

Additional Points

Are the actin levels equivalent between TDA and the mesenteric arteries? This is important as the mRNA data for both sets of genes are normalized to actin.

Please clarify what the “merge” label refers to in the legend for Figure 1.

Please indicate the temperature at which the experiments were performed.

Reaction rates are usually expressed as changes in concentration per unit of time. Please clarify whether the reaction rates presented in the manuscript correspond to rate constants.

Please indicate the wavelengths used to obtain the rate constants for the eNOS heme reduction experiments.

Author Response

Reviewer 3:

The kinetics of eNOS heme reduction by NADPH were reported by Abu-Soud and Stuehr in JBC 2000; 275(23)17349. Similar to Abu-Sound et al, the authors report that eNOS heme reduction follows biphasic kinetics. Please provide representative raw traces and corresponding fits in Figure 2.

Abu-Soud and Stuehr in JBC 2000 presented biphasic kinetics: one rate for the reduction of the flavin and one rate for the reduction of the heme. We have revised Figure 2 and have added an insert showing the kinetics and fit of the eNOS heme reduction. We reanalyzed our kinetics and found that if we used the same wavelengths as published by Abu-Soud and Stuehr in JBC 2000, the kinetic was monophasic. We have amended Table 1 accordingly.

Using a stopped-flow apparatus at 10oC, Abu-Sound et al reported a k = 85s-1 for the fast phase of eNOS heme reduction by NADPH. Does this correspond to the same process reported in Table 1 to occur with a rate constant of 0.02 s-1? Why is there such a large discrepancy?

The reduction rate of 85 s−1 reported by Abu-Soud et al. corresponds to the reduction of the flavin. Abu-Soud et al. reported the reduction rate of the heme rate constant as 0.005 s−1 at 10°C in the presence of L-arginine. Based on our new analysis, we report a reduction rate of 0.003 s−1 at 25°C in the presence of L-arginine. This small difference could be explained by differences in the quality of the protein and/or experimental variations.

Please provide the spectrum for fully reduced eNOS in Figure 2A to better show the isosbestic point at 410nm.

We have revised Figure 2 to show the fully reduced eNOS spectrum.

Consistent with Abu-Sound et al, the presence or absence of L-Arg is stated to have no effect on eNOS heme reduction rates. However, neither the rate constants with and without Arg, nor a statistical analysis of this comparison are provided.

This is an excellent point. We have revised Table 1 to show the reduction rates in the presence and absence of L-arginine. We measured an increase in reduction in the presence of L-arginine at 25°C (0.003 s−1 vs 0.008 s−1 in the absence of L-arginine). As discussed in our response to a previous question, the reduction rates are different as are the experimental conditions; therefore, we believe that statistical comparison of these rate constants in the presence or absence of L-arginine is not meaningful.

No information is provided regarding the number of replicates and standard deviations for the kinetic determinations, the r2 of the fits or any other statistical analysis.

We now acknowledge in the Methods section that every condition was duplicated (lines 174-175). We have included standard deviations in Table 1 and have included the r2 values with the fits of each of the kinetics in Figure 2, Figure 3, Supplemental Figure 1, Supplemental Figure 2 and Table 1.

The reduction rates for the reduction of alpha-globin by eNOS and by the CyB5R3/CyB5a system were reported by the authors in a 2018 paper in the Journal of Clinical Investigation. The only apparent difference between the original report and the present manuscript is that a previously unobserved rapid phase is now reported with a rate estimated at 0.5 s-1. Given that this observation seems to be the main difference between the 2018 paper and the present manuscript, the authors are advised to attempt a better characterization of this hyper-reductive rate by either working at lower temperatures or using a stopped-flow system.

Our previous study in which we measured the α-globin reduction rate by eNOS in the absence of the eNOS cofactors (BH4 and CaM/Ca2+) has been criticized (PMID: 30295642). In this study, we wanted to control for the functionality of the eNOS protein by measuring the reduction of one oxygenase domain of one monomer by the FMN domain of the other monomer in the functional eNOS dimer and to measure the reduction of α-globin in these conditions.

We initially obtained better simulations of eNOS reduction with two exponential rates by analyzing the variation in absorption in the Soret band corresponding to that close to the maximum and minimum of the steady-state differential spectrum: for instance, under anaerobic conditions, 390 minus 430 nm for the transition from ferric to ferrous heme. However, as reported by Abu-Soud et al., the analysis with a single absorption wavelength in the Soret, e.g., the maximum absorption of the steady-state spectrum of one species involved during the redox transition minus the absorption at 700-750 nm, was better simulated with one exponential rate. Note that the main kinetic rate obtained using a biexponential analysis was similar to that obtained with a monoexponential analysis. We have modified the rate constants in Table 1 based on these new analyses.

We are grateful for the reviewer’s suggestion to use a stopped-flow system, but we have been unable to do so because the quantity of protein required for this experiment was much higher than the amount of eNOS that we received from our collaborators. Also, it would have been difficult to perform these experiments under anaerobic conditions with our equipment.

The hyper-reductive rate is not apparent in figure 3A and the model used to fit the kinetic trace is not specified. Is this a biphasic or a monophasic model?

As discussed above, we reanalyzed our data and the kinetic trace shown is monophasic, using SigmaPlot to fit the raw values as a single exponential decay and corresponding to a reduction rate constant of 0.003 s−1 at 25°C. We have revised Table 1 accordingly.

Additional Points

Are the actin levels equivalent between TDA and the mesenteric arteries? This is important as the mRNA data for both sets of genes are normalized to actin.

The mean actin CT values are 25.07 ± 0.85 (SEM, n = 5) for TDAs and 24.20 ± 0.48 (SEM, n = 4) for mesenteric arteries. We believe these values to be comparable in the tissues analyzed.

Please clarify what the “merge” label refers to in the legend for Figure 1.

We have modified the legend of Figure 1 to clarify the merge images (line 196).

Please indicate the temperature at which the experiments were performed.

It was already specified in the Methods, section 2.6, that all enzymatic reactions were performed at 25°C. However, we have now also specified this in the legends for Figure 3 and Table 1.

Reaction rates are usually expressed as changes in concentration per unit of time. Please clarify whether the reaction rates presented in the manuscript correspond to rate constants.

We expressed all the rate constants in s−1 as we kept the stoichiometry of each partner at 1:1 based on the heme concentration and we did not vary the protein concentration. We could not use heme at a concentration of less than 1 µM if we wished to measure the Δ signals properly, and the limited amount of eNOS protein available to us meant that we could not increase the concentration and adjust our conditions to measure bimolecular rates. Bimolecular rates could be first estimated by divided the rate s-1 by 1 µM heme concentration.

Please indicate the wavelengths used to obtain the rate constants for the eNOS heme reduction experiments.

We have modified the axes of the figures so that they now show the wavelengths used to measure the rate constants.

Round 2

Reviewer 3 Report

The new dithionite traces show that there is no isosbestic point at 410nm thus calling into question the analysis shown in Figure 3. The rationale for now choosing a mono-exponetial versus the original biexponential fit is unclear. If the kinetics truly have a fast initial rate, it should be analyzed by stopped flow kinetics or at low temperature, otherwise it is just being ignored. If heme reduction rates by eNOS were originally reported as identical both in the presence and absence of L-arginine, it is unclear why the authors now state that these rates cannot be compared statistically. An n=2 is insufficient for analysis. If the objective of this manuscript was to compare the rates of alpha-globin heme reduction by eNOS in the presence and absence of critical cofactors (BH4 and CaM/Ca2+), then the authors should perform experiments in the presence and absence of such cofactors and compare them statistically to show if there is a difference. This manuscript still provides very limited new information/innovation and without sufficient rigor.

Author Response

Please find a cover letter in attachment.

Sincerely,

Christophe Lechauve
